# High-Density Lipoprotein Biomimetic Inorganic–Organic Composite Nanosystem for Atherosclerosis Therapy

**DOI:** 10.3390/polym17050625

**Published:** 2025-02-26

**Authors:** Yunpeng Zhang, Danni Liu, Yaoqi Wang, Qi Sun, Dong Mei, Xiaoling Wang, Yan Su, Siyu Liu, Chunying Cui, Shuang Zhang

**Affiliations:** 1School of Pharmaceutical Sciences, Capital Medical University, Beijing 100069, China; 13468311827@163.com (Y.Z.); dannyoldn@163.com (D.L.); wangyaoqi@ccmu.edu.cn (Y.W.); sunqi77@ccmu.edu.cn (Q.S.); liusiyu@mail.ccmu.edu.cn (S.L.); 2Engineering Research Center of Endogenous Prophylactic of Ministry of Education of China, Beijing Area Major Laboratory of Peptide and Small Molecular Drugs, Beijing Laboratory of Biomedical Materials, Beijing 100069, China; 3Laboratory for Clinical Medicine, Capital Medical University, Beijing 100069, China; meidong@bch.com.cn (D.M.); wangxiaoling@bch.com.cn (X.W.); suyanbch@sina.com (Y.S.); 4Department of Pharmacy, Beijing Children’s Hospital, Capital Medical University, National Center for Children’s Health, Beijing 100045, China; 5Medical Oncology Department, Pediatric Oncology Center, Beijing Children’s Hospital, Capital Medical University, National Center for Children’s Health, Beijing 100045, China

**Keywords:** cerium-manganese hybridized nanoparticles, ApoA1, atherosclerosis, antioxidant

## Abstract

Atherosclerosis (AS) is an important causative agent of cardiovascular diseases, and the occurrence and development of AS is accompanied by oxidative stress, so antioxidant therapy has become one of the strategies for the treatment of AS. This study aimed to design and construct an apolipoprotein ApoA1-modified inorganic–organic composite nanosystem for AS therapy, in which ApoA1 was modified onto carboxylated CeO_2_/Mn_3_O_4_ by covalent bonding, resulting in an inorganic–organic nanocomplex with a structure similar to that of high-density lipoprotein. The nanocomplex could effectively deliver the antioxidant nanoparticles to the AS plaque through the specific recognition between ApoA1 and the macrophage at the AS lesion site. For one thing, the nanocomplex could alleviate the oxidative stress environment of the AS site through the highly efficient antioxidant effect of CeO_2_/Mn_3_O_4_, which played a therapeutic role in the treatment of AS. For another, it could effectively eliminate the formed lipid plaques and maximally alleviate and treat AS by utilizing the cholesterol efflux effect of ApoA1.

## 1. Introduction

Cardiovascular diseases are sweeping the globe and have become the leading cause of death worldwide [1,2]. Atherosclerosis (AS), the leading cause of cardiovascular disease, is a chronic, systemic, and inflammatory disease that primarily affects the large- and medium-sized arteries [3]. Accumulation of lipids or fibrous substances in the arterial intima forming lipid plaques in the arterial wall leads to AS [4]. Nowadays, AS is characterized by a high morbidity and mortality rate, and changes in modern diet and lifestyle habits have led to a trend towards a younger population with this disease [5,6].

Studies have shown that the inflammation and intense oxidative environment of lesioned blood vessels are important triggers inducing the occurrence and development of AS [7,8,9]. Antioxidant and anti-inflammatory therapy and elimination of the inflammatory environment at the site of AS lesions have gradually become essential treatments for AS [10,11]. During the formation of atherosclerosis, oxidative stress caused by a high level of reactive oxygen species (ROS) is an important factor contributing to the development of inflammation in AS [12,13]. Therefore, alleviating the oxidative stress environment of AS lesion vessels through antioxidant therapy can effectively prevent the formation and further accumulation of AS plaques, which is of great value and significance for the treatment of AS.

Various antioxidant drugs, such as vitamin E, probucol, and coenzyme Q, have shown certain inhibitory effects on AS [14,15]. However, the shortcomings of the above drugs, such as their inability to be specifically distributed to AS lesions, short half-life, and low oxidative activity, have greatly limited their application in the clinical treatment of AS. Choosing the appropriate antioxidant is an important aspect of the successful application of antioxidant therapy in AS. Inorganic nano-based antioxidant nanoenzymes, such as cerium dioxide, manganese tetraoxide, vanadium pentoxide, etc., have gained widespread attention due to their simultaneous high catalytic activity of enzymes and excellent biological and physicochemical properties of nanomaterials [16,17]. Cerium dioxide nanoparticles (CeO_2_), as an excellent ROS-scavenging nano-agent, can catalyze the scavenging of superoxide, hydrogen peroxide, and hydroxyl radicals through the interconversion of the Ce^3+^ and Ce^4+^ ionic states, thus alleviating the oxidative stress environment at the lesion site [18,19]. Mn_3_O_4_ also has the potential to provide cytoprotection to the cells from oxidative damage [20]. The antioxidant and ROS scavenging ability of CeO_2_ can be further improved by the hybridization modification of CeO_2_ with manganese oxide to obtain a more efficient and stable antioxidant nanocatalyst (CeO_2_/Mn_3_O_4_) [21,22].

Inorganic nanomaterials have excellent optical and electrical properties and show unique advantages in bioimaging and disease treatment. However, inorganic nanomaterials have problems such as a poor biocompatibility, short blood half-life, and poor stability in aqueous systems, which prevent them from being widely used [23,24]. As with the majority of inorganic nanomaterials, CeO_2_/Mn_3_O_4_ has the disadvantages of a poor biocompatibility, easy metabolic clearance, poor dispersion in water-soluble media, and lack of tissue specificity, which greatly limits its application [25]. The rational and effective modification of inorganic nanomaterials are essential for their successful application. Inorganic–organic nanocomplexes obtained by combining inorganic and organic materials can effectively solve the above biological problems faced by inorganic nanomaterials in the process of drug delivery, and greatly expand the application scope of inorganic nanomaterials [26,27]. Currently, the commonly used method of organicization modification of inorganic nanomaterials is to prepare inorganic nanomaterials encapsulated by organic polymer compounds through chemical bonding or adsorption [24]. However, as exogenous compounds, organic polymers also face the challenges of a poor biocompatibility, susceptibility to immunization, and lack of targeting in the process of application. Therefore, choosing appropriate methods to modify inorganic nanomaterials and solving the biological problems in their application are the key to the successful application of inorganic nanomaterials.

High-density lipoprotein (HDL) is an endogenous lipoprotein with a particle size of 7–13 nm, and studies have shown that high levels of HDL are effective in preventing the development of AS and in eliminating lipid plaques that have formed in AS [28,29]. Apolipoprotein A1 (ApoA1) is an important component of HDL, and the interaction of ApoA1 with phospholipid molecules forms stable spherical or discoidal HDL particles [30]. ApoA1 can bind specifically to the ABCA1/G1 to enter macrophages and foam cells at the site of atherosclerotic plaques to facilitate the efflux of cholesteryl esters and other lipids from the cells that is known as reverse cholesterol transport (RCT) [31,32]. Thus, ApoA1-based mimetic HDL can not only be used as an excellent carrier for drug delivery to AS lesions, but also can play a role in AS plaque elimination by promoting cholesterol efflux and other pathways.

In this study, we designed a new type of bionic HDL nanoparticles with excellent antioxidant therapeutic effects (Figure 1). The inorganic–organic complex biomimetic nanoparticles (Apo-Ce/Mn) could not only utilize the targeting effect of ApoA1 on foam macrophages to effectively deliver the highly efficient antioxidant molecule CeO_2_/Mn_3_O_4_ to the AS lesion site, but also make use of the pro-lipid plaque elimination and exocytosis of ApoA1, through the synergism of both; it could inhibit the development of AS as much as possible, and eliminate the AS lipid plaque that has already formed, thus realizing the effective treatment of AS.

## 2. Materials and Methods

### 2.1. Materials

Cerium (III) nitrate hydrate (C916408, Macklin, Shanghai, China), manganese (II) chloride anhydrous (M8800, Solarbio, Beijing, China), oleic acid (O0011, TCI, Shanghai, China), oleylamine (O815176, Macklin, Shanghai, China), 1-Dodecanol (D807519, Macklin, Shanghai, China), xylenes (X820585, Macklin, Shanghai, China), DSPE-mPEG2k (S28722, Yuanye, Shanghai, China), and DSPE-mPEG2k-COOH (R2J0232521,xarxbio, Xian, China) were utilized in nanoparticle preparation. DIR (KeyGENBioTECH, Nanjing, China,) was employed to fluorescently label nanoparticles. The Apo-1 protein (FineTest, Wuhan, China) was used to modify nanoparticles as a targeting agent. Oil Red O (O104972, Aladdin, Shanghai, China) was employed for cell and aortic staining.

### 2.2. Synthesis of CeO_2_ Nanocrystals

A mixture consisting 0.43 g of cerium (III) nitrate, 2.7 g of oleylamine, and 0.03 g of oleic acid was prepared in 15 mL of 1-dodecanol. This mixture was heated to 120 °C under air. The solution was aged for a period of time until the solution turned yellow. During heating, 0.3 mL of ultrapure water was added when the reaction temperature reached 90 °C. Following the reaction, an excess of ethanol was introduced to facilitate the purification of the ceria nanocrystals. The nanocrystals were then collected via centrifugation.

### 2.3. Synthesis of CeO_2_/Mn_3_O_4_ Nanocrystals

A mixture consisting 0.09 g of as-synthesized CeO_2_ nanocrystals, 1.34 g of oleyl amine, 0.14 g of oleic acid, 0.26 mL of hydrochloric acid, and 15 mL of xylene was heated to 90 °C. Then 0.8 mL of 75 mg/mL fresh manganese (II) chloride solution was quickly added to the reaction solution. After 2 h aging, the solution was cooled down to room temperature, and the CeO_2_/Mn_3_O_4_ nanocrystals were washed with ethanol, and retrieved via centrifugation. The CeO_2_/Mn_3_O_4_ nanocrystals were well dispersed in chloroform. The relative Ce and Mn concentrations of the samples were estimated using inductively coupled plasma optical emission spectrometer (ICP-OES) (710-ES, VARIAN, Palo Alto, CA, USA).

### 2.4. Preparation and Characterization of Apo-Ce/Mn

The nanoparticles were prepared using an enhanced thin film hydration method. A mixture of 13.5 mg of DSPE-PEG-2k and 1.5 mg of DSPE-PEG-COOH-2k was added to 0.5 mL of CeO_2_/Mn_3_O_4_ in chloroform solution and the chloroform solution was replenished to 2 mL. The mixture was spun at 100 rpm for 20 min in order to evaporate the solvent. Subsequently, 1 mL of a 5% glucose solution was incorporated for hydration, followed by sonication to ensure proper dispersion. After sonication for 5 min, the solution was transferred to a 1.5 mL centrifuge tube and centrifuged at 10,000× *g* for 20 min (Centrifuge 5810-R, Eppendorf AG, Hamburg, Germany) at room temperature, and the supernatant was collected to obtain PEG-CeO_2_/Mn_3_O_4_ (PEG-Ce/Mn) nanoparticles. The 10 μg of Apo1 protein was added to 0.5 mL PEG-Ce/Mn nanoparticles followed by incubation at 4 °C for 24 h to prepare Apo-CeO_2_/Mn_3_O_4_ (Apo-Ce/Mn) nanoparticles. Purification and concentration of the nanoparticles were carried out using Amicon Ultra centrifugal filter units (MWCO-10,000, Millipore Inc., Billerica, MA, USA) through centrifugation for 10 min at 1400× *g* (Centrifuge 5810-R, Eppendorf AG, Hamburg, Germany). The resulting PEG-Ce/Mn and Apo-Ce/Mn complexes were concentrated and stored at 4 °C for future studies.

The relative Ce and Mn concentrations of the nanoparticles were estimated using ICP-OES. Cy5.5-labeled Apo1 protein was used to detect the loading efficiency of Apo1 protein. Cy5.5-Apo-Ce/Mn was prepared as previously outlined. The loading efficiency of Apo1 protein was analyzed using a fluorescence spectrophotometer (RF-6000, SHIMADZU, Kyoto, Japan). To achieve fluorescent labeling of PEG-Ce/Mn and Apo-Ce/Mn, 10 μL of 4.9 mg/mL DIR solution was added before spin distillation. Dynamic light scattering (DLS) and Zeta potential measurements were conducted using a Malvern Zetasizer (Nano Z90, Malvern, London, UK). The morphologies of PEG-Ce/Mn and Apo-Ce/Mn were observed using transmission electron microscopy (TEM) (JEM-2100, JEOL, Tokyo, Japan). Elemental valence analysis of Ce and Mn was performed using X-ray Photoelectron Spectroscopy (XPS) (Escalab 250XI, Thermo, Waltham, MA, USA).

### 2.5. Antioxidant Activity of Apo-Ce/Mn

The antioxidant activity within Apo-Ce/Mn was assessed by measuring the efficiency of H_2_O_2_ scavenging. In brief, Apo-Ce/Mn (with different concentrations of Ce) was introduced into a 0.5 mL H_2_O_2_ solution (500 mM) and allowed to react at 37 °C for 5 min. A 500 mM H_2_O_2_ solution without any treatment served as the control. The remaining H_2_O_2_ was quantified using a hydrogen peroxide detection kit (TO1076, Leagene, Beijing, China).

The clearance rate of H_2_O_2_ was calculated utilizing the following equation:Clearance rate of H_2_O_2_ (%) = (content of H_2_O_2_ with control − content of H_2_O_2_ with sample)/(content of H_2_O_2_ with control) × 100

The free radical scavenging activity of Apo-Ce/Mn was evaluated by scavenging ·DPPH. In brief, Apo-Ce/Mn (with different concentrations of Ce) was introduced into a 0.5 mL ·DPPH methanol solution (50 mg/mL) and allowed to react at 37 °C for 30 min. The absorbance at 560 nm was measured using a microplate reader (Multiskan Spectrum 1500, Thermo, Waltham, MA, USA).

The clearance of ·DPPH was calculated according to the following formula:Clearance rate of ·DPPH (%) = (OD control − OD sample)/(OD control) × 100

In addition, the ability of Apo-Ce/Mn to scavenge ·OH and ·DPPH radicals were examined using EPR (FA-300, JEOL, Tokyo, Japan).

### 2.6. Cell Culture

The RAW264.7 cell line was acquired from the Cell Bank of the Chinese Academy of Medical Sciences. For cell culture, DMEM (Dulbecco’s modified eagle medium) (Servicebio, Wuhan, China), fetal bovine serum (PAN-Seratech, Aidenbach, Germany), and penicillin and streptomycin (HyClone, Logan, UT, USA) were prepared. RAW264.7 cells were maintained in DMEM supplemented with 10% fetal bovine serum. All cell culture media were further supplemented with penicillin (100 U/mL) and streptomycin (100 μg/mL). Cultivation was carried out in a humidified 5% CO_2_ incubator at 37 °C.

### 2.7. Cell Toxicity of Apo-Ce/Mn

The toxicity of various concentrations of PEG-Ce/Mn and Apo-Ce/Mn formulations was evaluated by assessing cell viability through the MTT assay. Initially, RAW264.7 cells were plated into 96-well plates at a concentration of 5 × 10^3^ cells per well to adhere for 12 h. The cells were then treated with different concentrations of PEG-Ce/Mn and Apo-Ce/Mn (with a Ce dosage range of 0–30 μg/mL) for a period of 48 h. Post incubation, 25 μL of MTT solution (5 mg/mL, Aladdin, Shanghai, China) was added to each well, and the cells were incubated for an additional 4 h at 37 °C. Subsequently, 150 μL of DMSO (dimethyl sulfoxide) was added to each well, and the plate was agitated on a shaker for 5 min to ensure complete dissolution of the formazan crystals. Finally, the absorbance of each well was measured at 570 nm using a microplate reader (Multiskan Spectrum 1500, Thermo, Waltham, MA, USA) to determine the extent of cell viability.

Cellular viability was calculated using the formula:Cell viability (%) = (OD treatment group − OD blank)/(OD control group − OD blank) × 100.

### 2.8. Cellular Uptake of Apo-Ce/Mn

To investigate the internalization of PEG-Ce/Mn and Apo-Ce/Mn in macrophages and foam cells, RAW264.7 cells were seeded onto a 25 mm confocal dish at a concentration of 1 × 10^5^ cells per dish and allowed to adhere for 12 h. The induction of foam cell formation was achieved by treating the RAW264.7 cells with 25 μg/mL of oxidized low-density lipoprotein (Ox-LDL, IO1300, Solarbio, Beijing, China) for 48 h. Following this, both macrophages and foam cells were exposed to PEG-Ce/Mn and Apo-Ce/Mn labeled with DIR fluorescence for 4 h. Subsequently, after washing the cells thrice with PBS, the cultured cells were stained with Hoechst 33342 (Life Technologies, Carlsbad, CA, USA) and LysoTracker Green (Beyotime, Beijing, China). Observation and imaging were conducted using a laser confocal microscope (TCS SP5, Leica, Wetzlar, Germany). In conjunction with the imaging analysis, cellular uptake was also quantified using a BD LSRFortessa Flow Cytometer (BD, Franklin Lakes, NJ, USA) for flow cytometry analysis.

### 2.9. Scavenging ROS of Apo-Ce/Mn

RAW264.7 cells were seeded in 24-well plates at a concentration of 1 × 10^5^ cells per well and allowed to adhere for 12 h. Subsequently, the cells were stimulated with 20 μg/mL LPS (lipopolysaccharides, S11060, Yuanye, Shanghai, China) for 8 h. Following this, the cells were treated with various samples for an additional 12 h. After removing the drug-containing medium, the cells were washed with PBS, and a medium containing 10 μM DCFH-DA (HY-D0940, MCE, Monmouth Junction, NJ, USA) was added for co-incubation at 37 °C for 30 min. Then the cells were examined under an inverted fluorescence microscope (Ts2RFL, Nikon, Tokyo, Japan) to visualize the intracellular ROS levels. And the results were quantitatively analyzed using ImageJ software (v1.8.0, National Institutes of Health, Bethesda, MD, USA).

### 2.10. In Vitro Anti-Inflammation of Apo-Ce/Mn

RAW264.7 cells were seeded into 24-well plates at a concentration of 1 × 10^5^ cells per well and allowed to adhere for 12 h. The negative control group received fresh medium, while the experimental groups were stimulated with 20 μg/mL LPS for 8 h. Subsequently, cells were treated with different samples (30 μg/mL Ce) for 6 h and further incubated for an additional 42 h. Following this incubation period, enzyme-linked immunosorbent assay (ELISA) (PT512 for TNF-α, PI301 for IL-1β, PC125 for MCP-1, Beyotime, Shanghai, China) was utilized to assess typical inflammatory factors in the cell culture media supernatants, including tumor necrosis factor-α (TNF-α), interleukin-1β (IL-1β), and monocyte chemoattractant protein-1 (MCP-1). Moreover, the total protein content was determined using a BCA kit (Thermo, Waltham, MA, USA).

### 2.11. In Vitro Pro-Lipid Efflux of Apo-Ce/Mn

In short, RAW264.7 cells were seeded into 24-well plates at a concentration of 1 × 10^5^ cells per well and allowed to adhere for 12 h. The negative control group received a fresh medium, while the remaining groups were stimulated with 25 μg/mL Ox-LDL for 8 h. Following this, cells were treated with various samples for 6 h and then incubated for an additional 42 h. Afterward, the cells were washed with PBS, fixed using 4% paraformaldehyde, and subsequently stained with 0.3% ORO solution (dissolved in 60% isopropanol) for 20 min at room temperature. The ORO stain was then washed using PBS, and cell staining was observed under the bright field of a microscope (Ts2RFL, Nikon, Tokyo, Japan). And the results were quantitatively analyzed using ImageJ software (v1.8.0, National Institutes of Health, Bethesda, MD, USA).

### 2.12. Animals

Animal care and experiments were conducted in line with the Guide for the Care and Use of Laboratory Animals proposed by National Institutes of Health. All procedures and protocols were approved by the Institutional Animal Ethics Committee of Capital Medical University. Apolipoprotein E-deficient (ApoE^−/−^) mice, approximately 8 weeks old and weighing 20 g, were procured from the Animal Department of Capital Medical University (Beijing Laboratory Animal Center, Beijing, China). To establish the AS model, mice were fed a high-fat diet (HFD) containing 20% fat, 20% sugar, and 1.25% cholesterol for a duration of 8 weeks. Subsequently, the mice were divided into three groups randomly and received treatment via intravenous injection with 5% glucose solution, PEG-Ce/Mn and Apo-Ce/Mn (4 mg/kg Ce) every three days for six doses. After treatment, all mice were humanely euthanized through deep anesthesia followed by cervical dislocation to facilitate subsequent experiments.

### 2.13. In Vivo Plaque Targeting and Pharmacokinetics of Apo-Ce/Mn

To assess the targeting efficacy of Apo-Ce/Mn in AS model mice, DIR-Apo-Ce/Mn was employed for animal fluorescence imaging. The model mice received treatment via intravenous injection with free DIR-PEG-Ce/Mn or DIR-Apo-Ce/Mn for each mouse. Following an 8-h interval, the mice were placed under anesthesia and transcardially perfused with 30 mL PBS followed by 30 mL of 4% paraformaldehyde. Subsequently, the aorta and primary organs, including the heart, liver, spleen, lung, and kidney, were isolated and subjected to imaging using a fluorescence imaging system (IVIS Spectrum, PerkinElmer, Waltham, MA, USA). In addition, after administration, blood was collected from the tail vein at 2, 4, 8, and 24 h, and the aorta was taken 24 h later, and the blood and aorta were analyzed for Ce and Mn concentrations by ICP-OES.

### 2.14. In Vivo Pro-Lipid Efflux and Anti-Inflammatory of Apo-Ce/Mn

After the humane termination of the AS model mice, the aorta was meticulously excised, ensuring the complete removal of any adjacent adipose tissue. The excised aorta was then submerged in a 4% paraformaldehyde solution for fixation overnight. To quantify the plaque area, the aortas were stained with 0.3% ORO solution (dissolved in 60% isopropanol) for 30 min. Subsequently, the ORO stain was washed away using 70% ethanol, allowing for clearer visualization. In addition, ORO staining was performed on aortic tissue sections. The in vivo anti-inflammatory effects of Apo-Ce/Mn were examined using ELISA to determine immune factors such as MCP-1, IL-1β, and TNF-α in aortic tissue.

### 2.15. Immunohistology

The aortic tissue sections were analyzed immunohistologically for CD68, TNF-α, and IL-1β. Images of the stained sections were then observed using an automatic slice scanning system (Pannoramic scan, 3D HISTECH, Budapest, Hungary). And the results were quantitatively analyzed using ImageJ software (v1.8.0, National Institutes of Health, Bethesda, MD, USA).

### 2.16. In Vivo Safety Evaluation

The sections of major organs were subjected to Hematoxylin and Eosin (H&E) staining. Images of the stained sections were then observed using a fluorescence microscope (Eclipse Ti-SR, Nikon, Tokyo, Japan).

### 2.17. Statistics Analysis

The data were presented as mean ± SD (standard deviation). Statistical analyses were performed with GraphPad Prism (GraphPad Inc., v9.5, La Jolla, CA, USA) using one-way ANOVA. A value of *p* < 0.05 was deemed statistically significant, while *p* < 0.01 or *p* < 0.001 was considered highly significant.

## 3. Results

### 3.1. Characterization of Apo-Ce/Mn

Research has found that nanoparticles need to be less than 100 nm in diameter to accumulate in atherosclerotic plaques [33]. Meanwhile, HDL nanoparticles with a particle size of 10–30 nm accumulated better at the atherosclerotic plaque site than HDL nanoparticles with a diameter of 70 nm [34]. PEG-Ce/Mn and Apo-Ce/Mn were dispersed separately into PBS, and it was observed that the solutions were light brown in color and had obvious Tyndall effects (Appendix A). The particle size of the synthesized PEG-Ce/Mn was 35.45 ± 2.98 nm (Figure 2A) and the modified Apo-Ce/Mn was 38.72 ± 5.16 nm (Figure 2B) by DLS. The zeta potential of PEG-Ce/Mn was −25.27 ± 3.45 mV and that of Apo-Ce/Mn was −27 ± 3.45 mV, and the decrease in potential indicates the successful modification of Apo-Ce/Mn (Figure 2C). In addition, PEG-Ce/Mn and Apo-Ce/Mn were stored at 4 °C for 14 days, and their particle sizes and potentials did not change significantly, indicating good stability (Appendix A). The content of Ce and Mn in the nano-preparations was determined using ICP-OES, and the results are shown in Figure 2D. The contents of Ce and Mn in PEG-Ce/Mn were respectively 244.71 ± 9.47 μg/mL and 492.87 ± 12.67 ug/mL. And the Apo-Ce/Mn contained 236.32 ± 6.39 μg/mL of Ce and 421.05 ± 3.70 μg/mL of Mn. This indicated that the use of Apo1 to modify PEG-Ce/Mn did not affect the Ce and Mn content. As shown in Figure 2E, the synthesized CeO_2_/Mn_3_O_4_ was hexagonal in shape, consistent with previous reports, and this unique structure was shown to enhance its catalytic activity [21]. Both PEG-Ce/Mn and Apo-Ce/Mn nanoparticles were spherical and were around 30 nm in size. The ideal nano size of PEG-Ce/Mn and Apo-Ce/Mn contributed to the accumulation to the plaque site. Mixed valence states (Ce^3+^ and Ce^4+^) and their ability to switch between oxidation states play a crucial role in the scavenging of ROS [35]. At physiological pH, Ce^3+^ and Ce^4+^ revealed superoxide dismutase and catalase activities, respectively. And the ratio of Ce^4+^/Ce^3+^ is proportional to the cell viability, implying that higher Ce^4+^ content is less cytotoxic [36]. The elemental valence states in Apo-Ce/Mn were analyzed by XPS, with 79.44% Ce^4+^, 20.56% Ce^3+^, 59.79% Mn^2+^ and 40.20% Mn^3+^, which ensured the biosafety of Apo-Ce/Mn and provided the foundation for the antioxidant effect.

### 3.2. Cellular Uptake of Apo-Ce/Mn

In order to determine the concentration of cellular administration, MTT experiments were performed to examine the cytotoxicity of PEG-Ce/Mn and Apo-Ce/Mn, and it was found that at the maximum Ce concentration (30 μg/mL), the cell viability was still above 80% (Appendix A), indicating that PEG-Ce/Mn and Apo-Ce/Mn was not cytotoxic. Therefore, this concentration was used as the maximum dosing concentration in subsequent studies. From a targeted delivery perspective, ABCA1/G1 are abundant in macrophages, which are thought to be key receptors for ApoA1 binding [37]. Thus, ApoA1 has natural macrophage and foam cell targeting properties, which has led to the widespread use of ApoA1 in the development of anti-atherosclerotic biomimetic nanoparticles [38]. First, the uptake of different preparations by macrophages and foam cells was examined using CLSM. As shown in Figure 3A, the uptake of Apo-Ce/Mn was significantly greater than that of PEG-Ce/Mn in both macrophages and foam cells, and flow cytometry results (Figure 3B,C) similarly demonstrated that the cellular uptake efficiency of Apo-Ce/Mn was superior than that of PEG-Ce/Mn. However, it is worth noting that there was no significant difference in Apo-Ce/Mn uptake between macrophages and foam cells, which suggested that Apo-Ce/Mn could enter both macrophages to prevent their conversion to foam cells and foam cells to promote their lipid efflux.

### 3.3. In Vitro Antioxidant Effects of Apo-Ce/Mn

Atherosclerosis is associated with the production of bioactive oxLDL and enhanced oxidative stress. Various nanomedicines have been developed for antioxidant treatment of atherosclerosis and they are characterized by high free radical scavenging effects [39]. ·DPPH as a stabilizing free radical is often used for antioxidant capacity determination. As shown in Figure 4A, the color of the ·DPPH solution gradually became lighter with the increase of the concentration of Apo-Ce/Mn, indicating that Apo-Ce/Mn could remove ·DPPH. The clearance of ·DPPH could reach 41.59 ± 9.35% when the concentration of Apo-Ce/Mn was 5 μg/mL, and increasing the concentration to 30 μg/mL significantly increased the clearance to 73.18 ± 9.88%. Meanwhile, as shown in Figure 4B, oxygen bubbles generated by the decomposition of H_2_O_2_ could be clearly observed when the concentration of Ce reached 30 μg/mL with a clearance rate of 61.56 ± 13.02%, indicating that Apo-Ce/Mn had a catalytic effect. In addition, the results of ESR (Figure 4C) equally proved that Apo-Ce/Mn had the ability to scavenge ·DPPH and ·OH, affirming its antioxidant effect. Stimulation of RAW 264.7 macrophages with lipopolysaccharide (LPS) resulted in a state of oxidative stress and an increase in intracellular ROS levels. As shown in Figure 4D,E, the negative control group (NC) had lower ROS levels, while the LPS-treated positive control group (PC) had higher ROS levels, indicating that the oxidative stress cell model was successfully constructed. The ROS levels of the cells in the PEG-Ce/Mn and Apo-Ce/Mn treated groups decreased with the growing concentration of Ce, while the ROS levels in the Apo-Ce/Mn group were significantly lower than those in the PEG-Ce/Mn group, suggesting that Apo-Ce/Mn was more likely to enter the cells to exert antioxidant effects.

### 3.4. In Vitro Lipid Efflux and Anti-Inflammation of Apo-Ce/Mn

HDL removes excess cholesterol from foam cells in atherosclerotic lesions through a process called reverse cholesterol transport (RCT) [40]. The results of oil red O staining of foam cells are shown in Figure 5A. The higher concentration of Apo-Ce/Mn treatment resulted in a significant reduction of lipids in the cells relative to the PC group and was almost comparable to that of the NC group. PEG-Ce/Mn also had a pro-lipid efflux effect, which may be attributed to the effect of Ce/Mn, but was not as effective as that of Apo-Ce/Mn. The results of the semi-quantitative analyses also showed that the pro-lipid efflux effect of Apo-Ce/Mn was remarkable. This was because the ORO-positive area in the Apo-Ce/Mn group was reduced 8-fold compared to the control and 1-fold compared to the same concentration of PEG-Ce/Mn.

Additionally, large-scale clinical trial studies have found that inflammation is associated with the development of atherosclerosis and that anti-inflammatory interventions such as anti-cytokine therapies and colchicine have begun to show efficacy in the treatment of atherosclerosis, making anti-inflammation an effective strategy for the treatment of atherosclerosis [41]. ApoA1 also mediate anti-inflammatory effects by modulating Toll-like receptor 4-dependent signaling in macrophages [42]. Studies have also proved the cerium oxide nanoparticles have anti-inflammatory effects [35]. The levels of three inflammatory factors in the cell culture fluid were detected by ELISA (Figure 5C for IL-1β, Figure 5D for MCP-1, Figure 5E for TNF-α), and the levels of inflammatory factors in the Apo-Ce/Mn group were significantly lower compared with those in the PC group and the PEG-Ce/Mn group, suggesting that Apo-Ce/Mn had a stronger anti-inflammatory effect compared with PEG-Ce/Mn, which provided the basis for a better anti-atherosclerotic effect of Apo-Ce/Mn.

### 3.5. In Vivo Targeting Effect of Apo-Ce/Mn

The damaged vascular endothelium in plaques increases its permeability, allowing nanodrugs to penetrate through the vasculature into atherosclerotic lesions through enhanced permeability and retention (EPR) effects [43]. In addition, HDL biomimetic nanoformulations have the ability to efficiently target atherosclerotic plaques in vivo due to their natural targeting properties and ideal nano-size [44]. In vivo targeting experiments were performed using DIR-labeled PEG-Ce/Mn and Apo-Ce/Mn, with major organs and aortic vessels taken for imaging analysis after 8 h post tail vein injection. As seen in Figure 6A,B, among the major organs, the liver showed the strongest fluorescence signal, suggesting that PEG-Ce/Mn and Apo-Ce/Mn may be metabolized mainly by the liver. The results in Figure 6C,D illustrated that the fluorescence intensity in Apo-Ce/Mn was 1.07 times higher than that in PEG-Ce/Mn, so Apo-Ce/Mn could accumulate to the aortic arterial crown and had enhanced targeting properties compared with PEG-Ce/Mn. Also, the aortic Ce concentration detected by ICP-OES (Figure 6E) indicated that the Apo-Ce/Mn group has 1.58 times higher Ce concentration than the PEG-Ce/Mn group, which showed that Apo-Ce/Mn has excellent atherosclerotic plaque targeting properties. In addition, there was no significant difference in pharmacokinetics between Apo-Ce/Mn and PEG-Ce/Mn (Appendix A), as both PEG and Apo1 modifications enhanced the in vivo circulation time of the nanoparticles.

### 3.6. In Vivo Pharmacodynamic of Apo-Ce/Mn

Removal of atherosclerotic plaque is crucial for the treatment of atherosclerosis, and Apo1 has been shown to promote lipid efflux from foam cells via the RCT pathway, so high-density lipoprotein nanopreparations represented by Apo1 are considered to be used for precision medicine in atherosclerosis [44]. In vitro experiments have demonstrated that Apo-Ce/Mn promotes lipid efflux from foam cells. In in vivo experiments, overall oil red O staining of the aorta (Figure 6F) showed more positive areas (red) for lipid staining in the control and PEG-Ce/Mn groups and fewer positive areas in the Apo-Ce/Mn group. This suggested that Apo-Ce/Mn could similarly promote foam cell paper exocytosis in the plaque region in vivo. This was likewise demonstrated by the results of oil red O staining of aortic sections (Figure 6G), where the level of positivity was significantly lower in the Apo-Ce/Mn group than in the control and PEG-Ce/Mn groups.

Atherosclerosis is an inflammatory disease that is characterized by increased levels of several inflammatory factors, and thus anti-inflammatory therapy has become an effective strategy for the treatment of atherosclerosis [45]. The results of immunohistochemistry were shown in Figure 7A, CD68 represented the degree of macrophage recruitment, and the higher proportion of positive areas in the control group (64.11 ± 5.55) and the PEG-Ce/Mn group (55.66 ± 1.87) indicated that there was a large number of macrophage aggregation in the aorta, suggesting that the plaque has not been eliminated, whereas the positive ratio in the Apo-Ce/Mn group (39.85 ± 4.99) was significantly reduced, suggesting that Apo-Ce/Mn can be targeted to remove atherosclerotic plaques to achieve the therapeutic purpose. IL-1β, TNF-α and MCP-1 are inflammatory factors closely associated with the development of atherosclerosis. The results of immunohistochemistry (Figure 7A) and quantitative ELISA (Figure 7B,C) showed that the levels of IL-1β and TNF-α in the Apo-Ce/Mn group were significantly lower than those in the control group and the PEG-Ce/Mn group, and the results of ELISA (Figure 7D) showed that the level of MCP-1 was the lowest in the Apo-Ce/Mn group, which indicated that Apo-Ce/Mn had potent anti-inflammatory effects for the treatment of atherosclerosis.

### 3.7. In Vivo Safety of Apo-Ce/Mn

The in vivo safety of nanomedicine is crucial for their clinical application. The body weights of the mice in each group were continuously examined during the treatment period, and the results (Appendix A) proved that there was no difference in the body weights of the mice in each group. At the end of the treatment, HE staining of the major organs was performed, and the results (Appendix A) proved that the major organs were not damaged in all groups of mice. All these results indicated that PEG-Ce/Mn and Apo-Ce/Mn had a favorable in vivo safety profile.

## 4. Conclusions

In this study, we constructed an innovative inorganic–organic hybrid nanosystem, enhanced with apolipoprotein ApoA1, to address atherosclerosis (AS) from dual perspectives and optimize the clearance of lipid plaques. The system leverages the potent antioxidant properties of CeO_2_/Mn_3_O_4_(Ce/Mn) to mitigate the oxidative stress prevalent in AS-affected areas, thereby contributing to therapeutic outcomes. Concurrently, by harnessing the cholesterol efflux capabilities of ApoA1, it effectively removes established lipid plaques, thereby maximizing the mitigation and treatment of AS. Ce/Mn nanoparticles were successfully synthesized and the inorganic nanoparticles were modified using the organic materials PEG and Apo1 without affecting the effects of Ce/Mn. Ce/Mn had a strong antioxidant capacity, scavenging free radicals such as ·DPPH and ·OH in vitro, and treating AS in vivo through antioxidant effects. Apo1 promoted lipid efflux and exerts an anti-inflammatory effect in vitro, and had an in vivo ability to target AS plaques while eliminating them. This nanosystem combined antioxidant, anti-inflammatory, and pro-lipid efflux strategies to synergistically exert therapeutic effects on AS, and was the first time that Apo1 was used to modify Ce/Mn nanoparticles, which provided ideas for the development of novel inorganic–organic nanosystems.

## Figures and Tables

**Figure 1 polymers-17-00625-f001:**
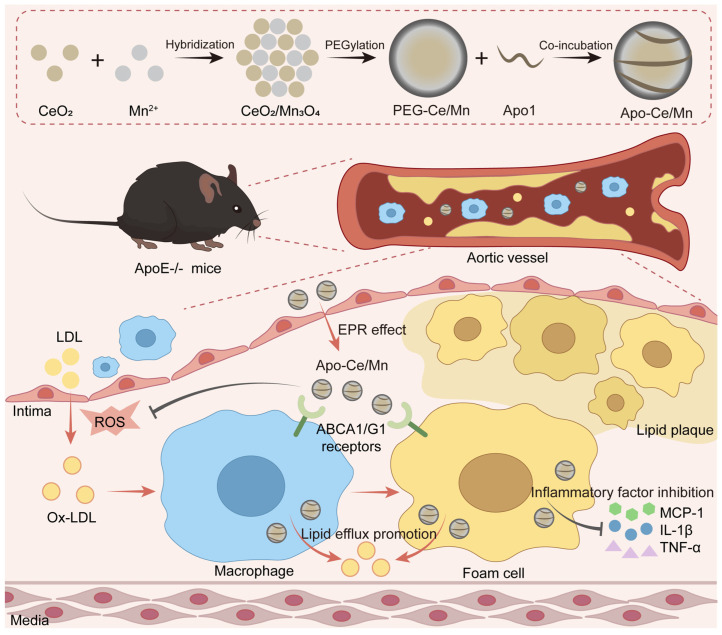
Synthesis and therapeutic mechanism of Apo-Ce/Mn. Apo-Ce/Mn can accumulate and penetrate into AS plaques; the Apo1 protein on the surface binds to ABCA1/G1 receptors to play a targeting role and promote lipid efflux, and CeO_2_/Mn_3_O_4_ hybrid nanoparticles can inhibit ROS generation and reduce inflammatory factor secretion, so Apo-Ce/Mn can target AS plaques and play a therapeutic role.

**Figure 2 polymers-17-00625-f002:**
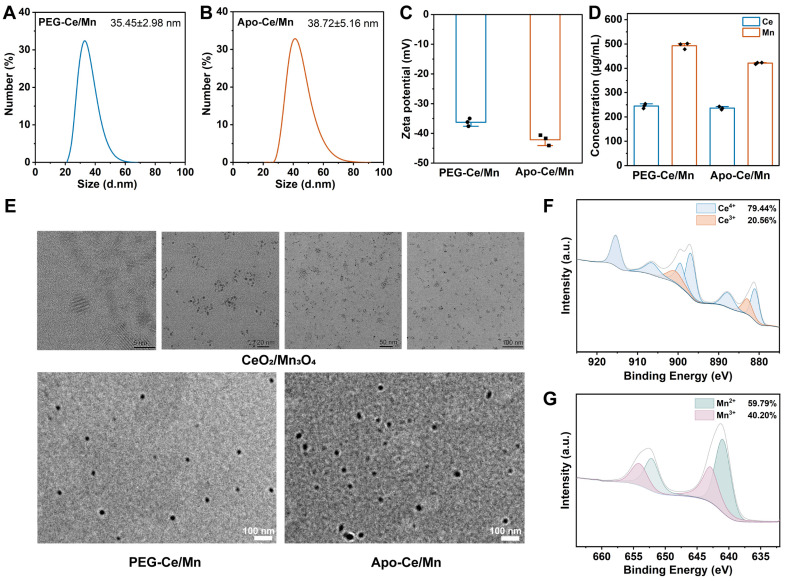
Characterization of PEG-Ce/Mn and Apo-Ce/Mn. (**A**) Particle size distribution of PEG-Ce/Mn. (**B**) Particle size distribution of Apo-Ce/Mn. (**C**) Zeta potential of PEG-Ce/Mn and Apo-Ce/Mn. (**D**) Ce and Mn content determined by ICP-OES. (**E**) TEM images of CeO_2_/Mn_3_O_4_, PEG-Ce/Mn, and Apo-Ce/Mn. (**F**) XPS analysis of Ce^3+^/Ce^4+^ content in Apo-Ce/Mn. (**G**) XPS analysis of Mn^2+^/Mn^3+^ content in Apo-Ce/Mn. n = 3.

**Figure 3 polymers-17-00625-f003:**
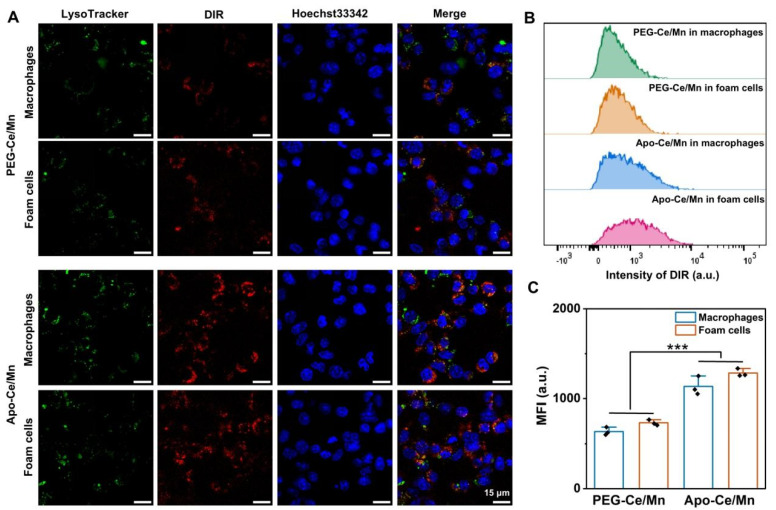
Cellular uptake in macrophages and foam cells. (**A**) Cell uptake of PEG-Ce/Mn and Apo-Ce/Mn in macrophages and foam cells for 4 h by CLSM; the lysosomes were labeled with LysoTraker (green), Apo-Ce/Mn was labeled with DIR (red) and the cell nucleus were labeled with Hoechst 33342 (blue). (**B**) Cell uptake of PEG-Ce/Mn and Apo-Ce/Mn in macrophages and foam cells for 4 h by FC. (**C**) MFI of Cy5 according to (**B**). *** *p* < 0.001, n = 3.

**Figure 4 polymers-17-00625-f004:**
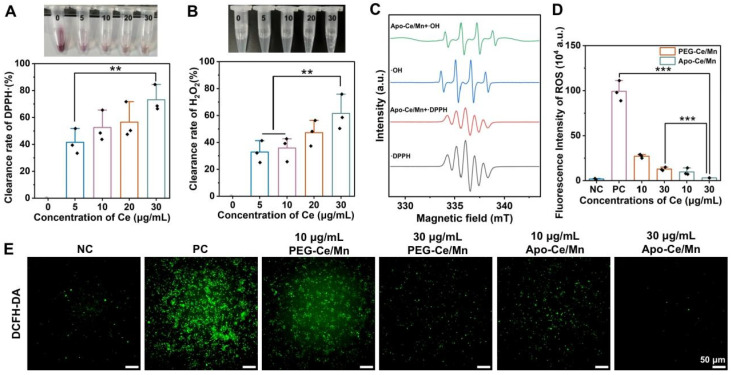
Antioxidant mechanism in vitro. (**A**) ·DPPH scavenging ability of Apo-Ce/Mn containing different concentrations of Ce. (**B**) The H_2_O_2_ scavenging efficiency of Apo-Ce/Mn containing different concentrations of Ce. (**C**) ·DPPH and ·OH scavenging by Apo-Ce/Mn detected by ESR. (**D**) Quantification of ROS fluorescence intensity according to (**E**) using ImageJ. (**E**) Detection of DCFH-DA reactive oxygen species in RAW264.7 cells. ** *p* < 0.01, *** *p* < 0.001, n = 3.

**Figure 5 polymers-17-00625-f005:**
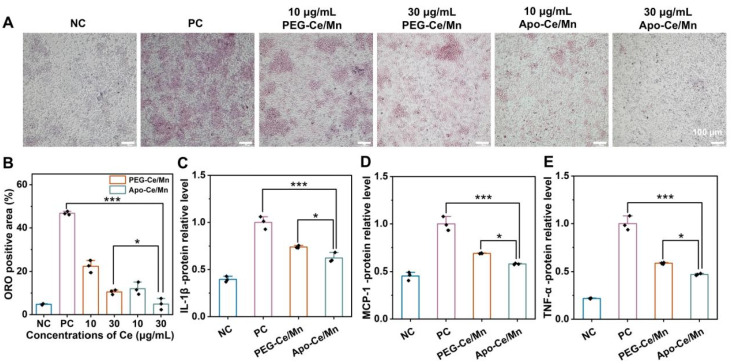
Promoting a lipid efflux mechanism and anti-inflammation in vitro. (**A**) ORO staining pictures of cellular lipids. (**B**) Analysis of positive area according to (**A**) using ImageJ. The levels of typical inflammatory cytokines such as IL-1β (**C**), MCP-1 (**D**), TNF-α (**E**) by ELISA. * *p* < 0.5, *** *p* < 0.001, n = 3.

**Figure 6 polymers-17-00625-f006:**
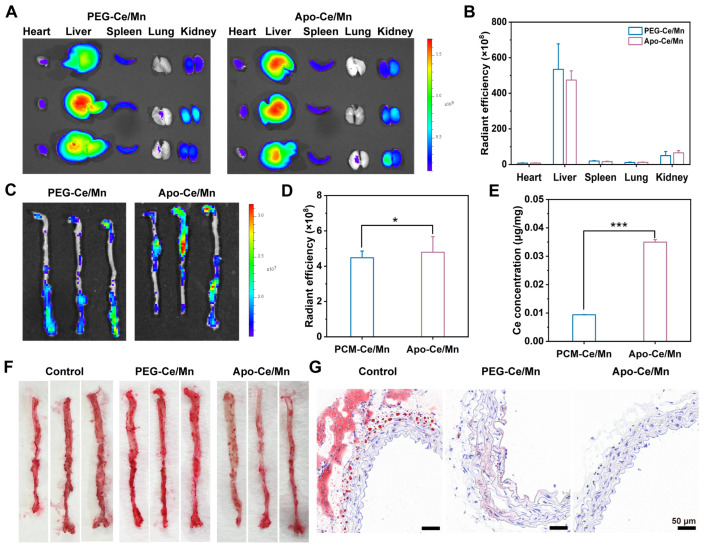
Targeting and pharmacodynamic evaluation in vivo. (**A**) Representative images of DIR fluorescent signals in major organs (spleen, lung, liver, and kidney). (**B**) Quantification of fluorescence intensity based on (**A**). (**C**) Representative images of DIR fluorescent signals in the aorta. (**D**) Quantification of fluorescence intensity based on (**C**). (**E**) Ce concentration in the aorta determined by ICP-OES. (**F**) Oil Red O staining of whole aortas. (**G**) Oil red O-stained images of aortic sections. * *p* < 0.5, *** *p* < 0.001, n = 3.

**Figure 7 polymers-17-00625-f007:**
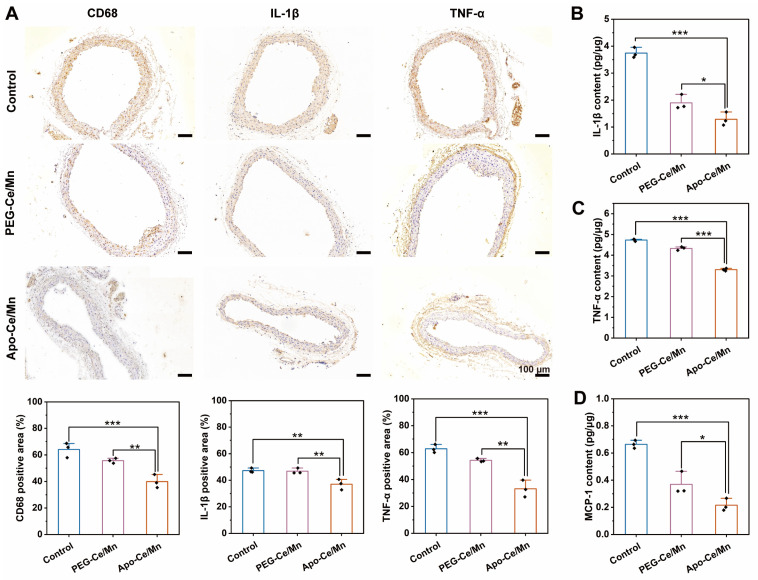
In vivo anti-inflammatory evaluation of Apo-Ce/Mn. (**A**) Immunohistochemistry (CD68, IL-1β and TNF-α) of aortic sections and semiquantitative analysis by ImageJ. (**B**) IL-1β contents in aortic tissue measured by ELISA. (**C**) TNF-α contents in aortic tissue measured by ELISA. (**D**) MCP-1 contents in aortic tissue measured by ELISA. * *p* < 0.5, ** *p* < 0.01, *** *p* < 0.001, n = 3.

## Data Availability

The original contributions presented in this study are included in the article/Appendix A. Further inquiries can be directed to the corresponding authors.

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
