# Peer review of "High-Density Lipoprotein Biomimetic Inorganic–Organic Composite Nanosystem for Atherosclerosis Therapy"

_polymers, 2025, doi:10.3390/polym17050625_

Round 1
Reviewer 1 Report
Comments and Suggestions for Authors
The manuscript describe the design and manufacture of biomimetic HDL to be used in the atherosclerosis treatment. The manuscript seems to me solid from the scientific point of view and brings new insights to the treatment of this pathology. Despite the fact that it focuses only animal trials, it is expected that may have interesting implications in the treatment of humans. I have some minor remarks to be considered by authors:
1) the pictures in Fig. 2 are small and difficult to be analyzed by readers. I recommend bigger pictures, particularly those that claim a hexagonal shape of the particle.
2) concerning the role of oxLDL in the atherosclerosis development, and its optical signature, authors could find in the literature some interesting papers, not cited in the present text.
In summary, the manuscript is interesting and I recommend its publication in Polymers, after minor revision.
In addition,
1) What is the main question addressed by the research? Authors describe the formation of a particle carrying drugs that may act on atherosclerotic plaques to reduce its size. 2) Do you consider the topic original or relevant to the field? Does it address a specific gap in the field? Please also explain why this is/ is not the case. Yes. All the efforts to reduce the size of these plaques are important for the therapy of atherosclerosis. The proposal of use CeO_2/Mn_3O_4 is very interesting and the results in mice seems relevant. 3) What does it add to the subject area compared with other published material? This type of approach seems to me new and the results are promising. 4) What specific improvements should the authors consider regarding the methodology? The use of biomimetic HDL particles is very interesting since the particles may be easily identified by the biological medium as something from its own, different from other manufactured nanoparticles. This makes its use more straightforward to therapies. 5) Are the conclusions consistent with the evidence and arguments presented and do they address the main question posed? Please also explain why this is/is not the case. The conclusions drawn in the manuscript are consistent with the initial proposal of the research. The experimental results (employing the methodology described in the text) leads to the conclusions. 6) Are the references appropriate? I suggest that authors look at references describing the in vitro oxidation of LDL and its nonlinear optical signal, since it could be useful to understand the role of oxidation of the HDL. 7) Any additional comments on the tables and figures. No additional comments.Author Response
Comments 1: the pictures in Fig. 2 are small and difficult to be analyzed by readers. I recommend bigger pictures, particularly those that claim a hexagonal shape of the particle.
Response 1: Thanks for the reviewer's comments, we have modified the images in Fig.2 (seen in the attachment named as "Reviewer 1"), the reason for the smaller TEM image of CeO2/Mn3O4 in Fig.2E is that CeO2/Mn3O4 itself had a very small particle size around 5~10 nm, so it would appear small when the magnification is changed.
Comments 2: concerning the role of oxLDL in the atherosclerosis development, and its optical signature, authors could find in the literature some interesting papers, not cited in the present text.
Response 2: Thanks for the suggestion. OxLDL is a very important part for the development of atherosclerosis. Our research confirmed that the Apo-Ce/Mn nanoparticles had a strong antioxidant property, which will prevent the formation of oxLDL. However, considering the study focused on the treatment of atherosclerosis rather than the role of oxLDL in the progression of AS, so we did not concern the role of oxLDL in our paper.

Reviewer 2 Report
Comments and Suggestions for Authors
In the manuscript entitled „High-density lipoprotein biomimetic inorganic organic composite nanosystem for Atheroslerosis therapy” the novel nanoparticles efficiently fighting atherosclerosis. The elaborated material acts in two ways, namely, mitigate the oxidative stress environment and removes lipid plaques from arterial walls. According Authors and according to the results of biological studies conducted in vitro and in vivo, the developed combined inorganic-organic system proved to be a very effective therapeutic agent. The whole manuscript is prepared very clearly and the sections are very consequently described. The descriptions of the experiments are prepared with care. However, it should be noted that for the most part of the manuscript, the results of biological studies of the acting of an analyzed material on living cells and organisms are discussed. Description of the obtained therapeutic complex material is rather modest.
Some minor remarks are following:
1) The authors should reconsider the size of the drawings and pictures, which should generally be enlarged to better follow the details;
2) The dispersity of the nanoparticles is missing (usually is given with the particles diameter obtained from DLS measurements).
3) The Ce and Mn contents in nanoparticles should be given in micrograms per units of weight of the nanoparticles.
4) Section 2.4: how is bound Apo1 on the surface of hydrophilic (containing PEG) PEG-Ce/Mn nanoparticles? Whether the effect of desorption or washing out of Apo1 from the nanoparticles is not observed?
5) Are there suggestions of mechanism of degradation of nanoparticles (decay) after their uptake by cells?
6) Figure 2C. Why do nanoparticles containing PEG on the surface (PEG-Ce/Mn) have a relatively large negative charge?
7) Figure 2 caption: description of TEM images is point E) instead of D).
Author Response
Comments 1: The authors should reconsider the size of the drawings and pictures, which should generally be enlarged to better follow the details;
Response 1: Thanks to the reviewer's comments, we have modified some of the image sizes to make them clearer.
Comments 2: The dispersity of the nanoparticles is missing (usually is given with the particles diameter obtained from DLS measurements).
Response 2: In the DLS determination, due to the small nanoparticle size, the dispersion determination was susceptible to impurities in the measuring cup, and therefore the PDI obtained from the DLS determination was not referenced.
Comments 3: The Ce and Mn contents in nanoparticles should be given in micrograms per units of weight of the nanoparticles.
Response 3: The Ce and Mn contents in the manuscript were presented as ug/ml in 3.1. Characterization of Apo-Ce/Mn because the Ce and Mn contents in PEG-Ce/Mn and Apo-Ce/Mn solutions were determined, and considering the large error in determining the weight of the nanoparticles after freeze-drying, no freeze-drying was carried out to obtain the Ce and Mn contents per unit weight.
Comments 4: Section 2.4: how is bound Apo1 on the surface of hydrophilic (containing PEG) PEG-Ce/Mn nanoparticles? Whether the effect of desorption or washing out of Apo1 from the nanoparticles is not observed?
Response 4: In 2.4. Preparation and Characterization of Apo-Ce/Mn, DSPE-PEG-COOH-2k was used to synthesize PEG-Ce/Mn. DSPE-PEG-COOH-2k is a PEG material containing an activated carboxylate group that interacts with amino-containing proteins, and thus the Apo1 amino group can interact with this activated carboxyl group and thus link to the PEG-Ce/Mn. During the preparation process, an excess of Apo1 was added to ensure that Apo1 was linked to DSPE-PEG-COOH-2k as much as possible, and Apo1 was labeled with Cy5.5 for fluorescence quantification, which showed that the linkage efficiency of Apo1 was 77.21 ± 4.10%.
Comments 5: Are there suggestions of mechanism of degradation of nanoparticles (decay) after their uptake by cells?
Response 5: After a literature search, no cellular degradation (decay) studies on CeO2/Mn3O4 nanoparticles or HDLized nanoparticles were found. In addition, PEG is a neutral, non-toxic polymer with unique physicochemical properties and good biocompatibility, and one of the very few synthetic polymers approved by the FDA for in vivo injectable medicinal use. The cellular degradation (decay) of PEG has not been studied in detail. The animal experiments in this study showed that the nanodelivery system had a good in vivo safety profile; therefore, we believe that cellular degradation (decay) studies were unnecessary.
Comments 6: Figure 2C. Why do nanoparticles containing PEG on the surface (PEG-Ce/Mn) have a relatively large negative charge?
Response 6: In 2.4. Preparation and Characterization of Apo-Ce/Mn, DSPE-PEG-COOH-2k was used to synthesize PEG-Ce/Mn. DSPE-PEG-COOH-2k is a PEG material containing an activated carboxylate group, The reactive carboxyl group will dissociate in PBS solution and exhibit negative electronegativity; thus PEG-Ce/Mn has a relatively large negative charge.
Comments 7: Figure 2 caption: description of TEM images is point E) instead of D).
Response 7: Thank the reviewer for the suggestion, we have revised it in the manuscript.
